# Oral Glucose Tolerance Testing Using Candy: A Sweet Solution to Improve Screening in Children with Cystic Fibrosis?

**DOI:** 10.3390/children10081317

**Published:** 2023-07-31

**Authors:** Caroline Weeks, Sarah Jackson, Nadir Demirel, Janelle Olson, Vicki Dean, Caitlin Pyrz, Ana L. Creo

**Affiliations:** 1Mayo Clinic Pediatric Endocrinology Cystic Fibrosis Center, Mayo Clinic, Rochester, MN 55905, USA; carolinecweeks@gmail.com (C.W.);; 2Division of Pediatric and Adolescent Medicine, Mayo Clinic, Rochester, MN 55905, USA; 3Division of Pediatric Pulmonology, Rochester, MN 55905, USA

**Keywords:** glucose tolerance, cystic fibrosis, cystic fibrosis-related diabetes

## Abstract

Introduction: Oral glucose tolerance testing is recommended for all children with CF older than 9 years, yet compliance remains poor across centers. Methods: We performed a small pilot study assessing the glycemic curves and participant satisfaction in seven children and adolescents. Results: We chose a dextrose-based candy (Nerds^®^) free of any fat, fiber, gelatin, or corn syrup and performed the candy OGTT 1–4 days following the standard oral dextrose solution OGTT. Glucose values at 120 min were similar between the candy and oral dextrose solution (*p* = 0.8986). Conclusions: Our small pilot suggests that a carefully selected candy alternative may result in similar glycemic OGTT when compared to the standard oral dextrose solution. However, some participants preferred the oral dextrose solution to candy due to having to consume a large volume in a short period of time. This may have significant implications as centers consider candy alternatives to increase OGTT adherence rates.

## 1. Introduction

The Cystic Fibrosis (CF) Foundation recommends yearly screens for diabetes for all people with CF ages 10 and older via a 2 h oral glucose tolerance test (OGTT) [1]. Current adherence rates to this test are poor across many centers, with minimal published data describing solutions [2,3]. For example, in 2017, only 43% of patients aged 10 or older completed the test among a wide network of accredited CF centers [3]. While there are many reasons why OGTT screening rates may be low, the taste of the glucose solution used for the OGTT may be a significant deterrent in the pediatric CF population. Recently, Racusin et al. explored using candy to replace oral dextrose solutions in pregnant women with good results, but this has not been well-described in the CF population [4,5,6]. We aimed to perform a small pilot study to determine if a dextrose-based candy (Nerds^®^ candy, Chicago, IL, USA) would yield comparable glycemic curves to the traditional oral dextrose solution used for the OGTT. Second, we aimed to describe the participant’s overall satisfaction with the candy substitution as compared to the use of the standard dextrose liquid solution.

## 2. Methods

We recruited 10 participants with CF ages 10 to 21 years old between January and August 2021. We excluded participants with a CF flare requiring additional medicines or recent hospitalization, CF-related diabetes diagnosis, steroid use, inability to tolerate oral feeds, prior intestinal surgery, or concern for allergy or intolerance to Nerds^®^ candy. Nerds^®^ candy was chosen as it has no added fat, fiber, pectin, gelatin, or ingredients to which children have common intolerances such as gluten or high fructose corn syrup. Participants unable to finish the study in its entirety due to lack of follow up were also excluded. Participants first underwent the traditional clinical OGTT using the oral dextrose solution of 1.75 g/kg or a maximum of 75 g sugar. Participants had 5 min or fewer to consume the oral dextrose solution. Point-of-care blood sugars via fingerstick reading were obtained at 0 and 120 min. A second OGTT using the candy (Nerds^®^ candy) with the same protocol and weight-based dose was performed ±7 days from the traditional OGTT. Glucose excursions were compared between the standard and candy OGTT. Participants filled out a 5-question survey regarding their experience and satisfaction using the substitution of oral dextrose drink compared with the candy in the OGTT. The study was approved by the Mayo Clinic Institutional Review Board 20-002446 and registered trial NCT04579939. We used the Shapiro–Wilk test to probe normality in the OGTT for both the oral dextrose drink and Nerd’s treatment. The effect of the substitution of oral dextrose drink with Nerds^®^ candy on the OGTT was analyzed using a two-tailed paired t-test. Frequency and percentage were used to describe the survey results. Analyses were made in Prism 9 software version 9.2.0 (332).

## 3. Results

Out of the ten participants recruited, seven were able to complete the study. Initial recruitment was overall limited due to the COVID-19 pandemic, and completion of both arms of the study was not possible for three participants either due to last-minute health or exposure concerns. We included seven participants, most of whom were males *n* = 5 (71.43%). The median age was 16 years (R: 12–19). The Shapiro–Wilk test applied to both groups during minute 0 (OGTT with oral dextrose drink and Nerds^®^ candy) yielded a *p*-value = 0.8965 and 0.6647, respectively, proving that there is a normal distribution in the sample despite including few participants. When comparing the performance of both treatments, the average difference in blood glucose between Nerds^®^ candy and oral dextrose drink at minute 0 was −1.43 mg/dL, *p* = 0.5307 (95%CI −6.684 to 3.826); the distribution is shown in Figure 1A. At 120 min, the average difference in blood glucose between Nerds^®^ candy and oral dextrose drink was 2.71 mg/dL, *p* = 0.8986 (95%CI −47.25 to 52.68); the distribution is shown in Figure 1B. Regarding participant answers to the survey on experience, two participants (28.57%) were “unsatisfied” with the substitution of oral dextrose drink with Nerds^®^ candy in the OGTT, and two (28.57%) stated they felt satisfied. The remaining participants had neutral responses indicating that they would not mind if future tests were conducted with Nerds^®^ candy or oral dextrose solution (Figure 2, Table 1). Four participants indicated that they would not likely be able to complete the OGTT with Nerds^®^ candy again (Table 1). As a final question, the majority of participants (57.14%) replied that oral dextrose solution was preferred over Nerds^®^ candy.

## 4. Discussion

Recently, significant scientific advancements within the CF community in the realm of modulator therapies are effectively causing patients with CF to live longer, and it is projected that over 50% of patients are expected to develop CFRD in their lifetime [7]. This chronic comorbidity superimposed on an already burdensome pulmonary diagnosis poses significant challenges for not only patients with CF but also their medical providers. Recent studies and online surveys have been conducted with the aim of better understanding the patient perspective on developing and managing CFRD. It was found that despite information being available to patients in clinics, significant knowledge gaps about the condition still existed. As many as 40% of adolescents did not know how CFRD was different from other types of diabetes mellitus or that it shared characteristics of both type 1 and 2 diabetes nor what the pathophysiology was to cause the condition [8,9]. It is also a shared belief of patients with CF that care teams could improve the level of education and mental preparation imparted to patients for the lifelong daily management of CFRD, including daily injections, blood glucose monitoring, and potential carbohydrate counting [10]. One is left to assume that perhaps the earlier introduction of CFRD-related education to families is important to improve overall buy-in and adherence to treatments.

In addition to educational gaps, other current challenges facing the CF community are both access to specialty diabetes care and continuity of practice across centers. A recent Canadian-based survey found that fewer than 25% of patients had access to an endocrinologist with both expertise and interest in CFRD. Additionally, the heterogeneity in approaches to care and screening methods for CFRD across various centers further complicates the landscape. For example, guidelines recommend the use of the OGTT starting at 10 years of age; however, some centers conduct this test earlier, and some use different methods altogether, such as the use of continuous glucose monitoring [7]. Many patients diagnosed with CF at an early age follow their pediatric pulmonologist and CF team and are later forced to transition to a separate adult-based clinic which may have different approaches to clinic visits and overall pulmonary care than what the patient is used to. A recent program called CF R.I.S.E. was introduced by the CF Foundation and served as a standardized toolkit to equip the adolescent with CF to become more independent with their CF-related milestones and medical care so that they can better advocate for themselves and foster a smoother transition to the adult CF center environment [11].

As mentioned earlier in this article, data and clinical guidelines support the use of the OGTT for screening of CFRD, yet adherence rates are far lower than desired. In 2019, only 67% of eligible children and 37% of adults underwent screening [12]. Various reasons for avoiding this screening have been noted, such as fear of needles, lack of classic diabetes-related symptoms, burden of time, or, in the case of our particular CF center, the distaste for the standard liquid glucose solution [13].

While the number of study participants in our study is very small and much larger samples are needed to make definitive statements, our pilot data suggest that using an OGTT candy may produce similar screening test results to traditional oral dextrose OGTT. However, despite adequate test results using candy for the OGTT, some participants preferred the non-candy oral dextrose solution, and neither the glucose solution nor the candy alternative was superior. This may be because all participants were adolescents, there was less appeal for eating candy in older teens, and younger children may have different perceptions. As studies have shown, age and stage of development have a correlation to food choice preferences, and perception may play a larger role than realized [14]. It is also possible that we did not have a large enough sample to truly make an assessment if using candy was a further deterrent to completing OGTT. Previous studies have evaluated alternatives such as jellybeans, candy twists, and juice [4]; however, to our knowledge, our pilot study is the first to demonstrate the effect of our chosen candy alternative (void of added ingredients) on glucose excursion in participants with CF or any other condition requiring OGTT screening. The hard, crunchy nature of the Nerds^®^ candy may have been a dissatisfier, and a different candy may produce different results. While many pediatric providers may find the appeal of using a candy alternative to increase CFRD screening, this approach may not be enough to improve screening adherence. Other United States-based CF centers have recently published improved adherence via formal quality improvement initiatives [15] utilizing the implementation of pre-clinic educational mailings and post-clinic reminder phone calls [16]; however, no centers have shown a combination of these strategies with the addition of substituting ODS (oral dextrose solution) alternatives. In addition to further research aimed at standardizing alternative screening methods, such as continuous glucose monitoring, further work demonstrating creative solutions to address suboptimal adherence to continuing to increase screening rates for CFRD is needed, given the critical importance of detecting and treating to improve mortality outcomes in all people living with CF.

## 5. Study Limitations

Our study had several significant limitations, which must be considered in light of our results. Most importantly, the sample size is extremely small. As the study was performed during the peak of the pandemic, this greatly influenced the number of patients with cystic fibrosis (considered high risk for severe COVID infection) we were able to accrue. As such, we only present this study as a potential novel idea that would need much more rigorous testing prior to being used in clinical practice. Secondly, we chose a candy that was dextrose for optimal comparability. If this study were to be replicated outside of the United States, where our specific candy was not available, alternatives could be considered, but important considerations would be choosing a local candy that does not contain ingredients prone to allergy or intolerance, such as high fructose corn syrup, gluten, or gelatin products. The candy choice should have no fat whatsoever and no protein. In addition, the candy should not have any added types of fibers, such as pectin. Lastly, a candy should be able to be in an easily weighable form to allow for careful weight-based dosing. Other uniquely important factors that our study was unable to capture yet should not be ignored include both socioeconomic status and the impact that parenting styles and family dynamics play on overall food and eating behaviors. Out of the entire cohort, three patients were receiving Medicaid (a United States income-based government-assisted medical insurance program), and the remainder were privately insured. We, unfortunately, do not know information about our patients’ mealtime patterns at home (e.g., do families eat around a dinner table together, do they all aid in meal preparation, etc.), but it would be an interesting element to evaluate in future studies. As varied research has underscored, income, level of education, access to food, and other factors, such as stability of home life, affect total population health [17,18].

Lastly, on a technical note, the blood samples were performed from point-of-care devices, which in themselves have some variability in glucose values. However, in this population, designing a study with blood draws for serum glucose would have further decreased likely enrollment and accrual.

## 6. Conclusions

Our small pilot suggests that a carefully selected candy alternative may produce comparable glycemic OGTT results to the traditional oral dextrose solution. However, some participants preferred the oral dextrose solution to candy due to having to consume a large volume in a short period of time. This may have significant implications as other centers consider candy alternatives to increase OGTT adherence rates.

## Figures and Tables

**Figure 1 children-10-01317-f001:**
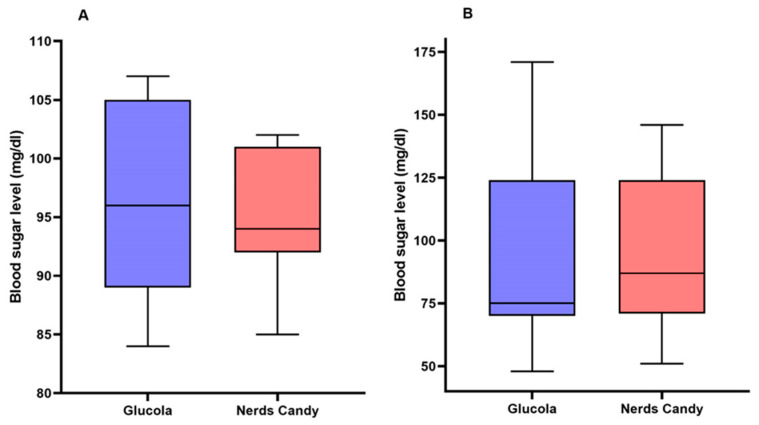
Sample distribution for blood sugar levels for Glucola and Nerds^®^ candy at (**A**) minute 0 and (**B**) minute 120.

**Figure 2 children-10-01317-f002:**
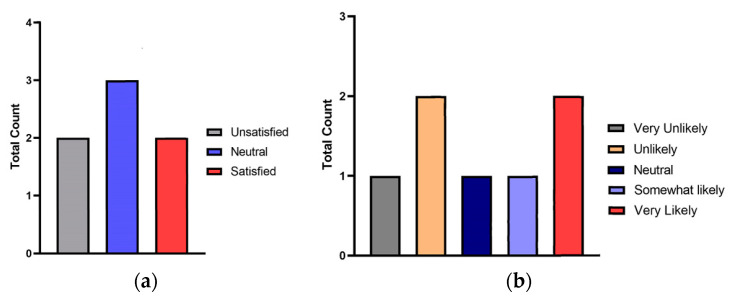
Summary of patients’ answers to the key survey questions: (**a**) “On a scale of 1–5, how satisfied were you with the Oral Glucose Tolerance Test (OGTT) using the NERDS candy?”; (**b**) “On a scale of 1–5, what is the likelihood of you completing the OGTT with NERDS candy again if asked?”.

**Table 1 children-10-01317-t001:** Participant Survey Results (*n* = 7).

	Mean, Mode (IQR)
**Question 1** On a scale of 1–5 (1 being not at all satisfied and 5 being very satisfied) how satisfied were you with the Oral Glucose Tolerance Test (OGTT) using the Nerds^®^ candy?	3, 2–4 (2)
**Question 2 ** Did you experience any unpleasant symptoms after the OGTT with Nerds^®^ candy?	0, 0–1 (0)
**Question 3** What symptoms did you experience?	Mild Fatigue *n* = 1 No Symptoms *n* = 6
**Question 4** On a Scale of 1–5 (1 being not at all likely and 5 being very likely) how likely would you be to complete a future OGTT with Nerds^®^ candy?	3, 1–5 (3)
**Question 5** If you could complete your next OGTT with Nerds or oral dextrose solution, which option would you prefer?	Glucola *n* = 4 Nerds^®^ *n* = 1 No Preference *n* = 2

## Data Availability

The data presented in this study are available on request from the corresponding author. The data are not publicly available due to privacy reasons.

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
