# Peer review of "Oral Glucose Tolerance Testing Using Candy: A Sweet Solution to Improve Screening in Children with Cystic Fibrosis?"

_children, 2023, doi:10.3390/children10081317_

Round 1

Reviewer 1 Report

Data seems to show that candy versus oral glucose are equivocal not that glucose is superior. 2 preferred glucose, 2 preferred nerds, 3 were indifferent - so not sure how that make glucose superior? 

Discussion and conclusion need to be corrected to say neither was superior. And glucose results were definitely similar. 

Also ODS was not defined - please define.

Author Response

Data seems to show that candy versus oral glucose are equivocal not that glucose is superior. 2 preferred glucose, 2 preferred nerds, 3 were indifferent - so not sure how that make glucose superior? 

Discussion and conclusion need to be corrected to say neither was superior. And glucose results were definitely similar. 

Also ODS was not defined - please define.

Reply:

This is a great point, we agree that this discussion and conclusion should be corrected to point out that neither glucose solution or Nerds candy were superior. The manuscript from lines 138-140 now reads, “However, despite adequate test results using candy for the OGTT, some participants preferred the non-candy oral dextrose solution and neither the glucose solution or candy alternative were superior.”

ODS should be defined as oral dextrose solution; the correction has been made in line 156.

Reviewer 2 Report

The brief report “Oral Glucose Tolerance Testing Using Candy: A Sweet solution to Improve Screening in Children with Cystic Fibrosis?” investigates the usefulness of a dextrose-based candy (Nerds®) free of any fat, fiber, gelatin, or corn syrup as an alternative to dextrose solution for the oral glucose tolerance test (OGTT). The reason for testing an alternative is the poor adherence rate for yearly diabetes screening for people with cystic fibrosis. Despite the small number of included patients, 7 people between 12 and 19 and uneven gender distribution (5 males), the results reported are interesting as candy poses an alternative to offer for people who find that the dextrose solution less appealing. This report is of general interest, the figures are of good quality, the references are appropriate and the statistical analysis is done according to proper practices. As the authors point out, larger cohorts are needed to draw any definite conclusions. There is only one concern with the conclusion:

For example, line 19:

“most participants preferred the oral dextrose solution to candy”

This conclusion does not strictly adhere to the results, because 2 people would very likely complete another OGTT with NERDS candy, 1 person somewhat likely and 1 person was neutral. In addition, only two out of seven people were averse to using NERDS again. These results do not indicate that the participants preferred oral dextrose to candy. Instead, the results say that patients did not prefer NERDS to the oral dextrose solution. The text should be changed accordingly.

Author Response

For example, line 19:

“most participants preferred the oral dextrose solution to candy”

This conclusion does not strictly adhere to the results, because 2 people would very likely complete another OGTT with NERDS candy, 1 person somewhat likely and 1 person was neutral. In addition, only two out of seven people were averse to using NERDS again. These results do not indicate that the participants preferred oral dextrose to candy. Instead, the results say that patients did not prefer NERDS to the oral dextrose solution. The text should be changed accordingly.

Reply:

Agree that this conclusion should be edited to stating that patients did not prefer Nerds candy to the oral dextrose solution. Lines 19, 138, and 190 have also been changed to “some participants preferred the oral dextrose solution to candy.”

Reviewer 3 Report

Dear colleagues,

Thank you for your research efforts performed in a very difficult moment for the whole medical community. I can understand the reason and particular moment in time that were generating such a small sample of patients in this pilot study.

Families have a significant impact on adolescent perception and behavior related to food. Several important aspects can be revealed by a more detailed evaluation of families of these children.

1. A potential bias factor can be represented by socio-economic status, because such adolescents are presenting persistently unhealthier dietary profiles [Fernández-Alvira JM, et al. Prospective associations between socio-economic status and dietary patterns in European children: the Identification and Prevention of Dietary- and Lifestyle-induced Health Effects in Children and Infants (IDEFICS) Study. Br J Nutr. 2015 Feb 14;113(3):517-25. doi: 10.1017/S0007114514003663.].

Could you evaluate socio-economic distribution patterns and correlation in your small group? Are there any discrepancies in perception related to this aspect?

2. Adolescents have infrequently healthy eating habits, and this moment in time is a difficult one, during childhood [Beal T, Morris SS, Tumilowicz A. Global Patterns of Adolescent Fruit, Vegetable, Carbonated Soft Drink, and Fast-Food Consumption: A Meta-Analysis of Global School-Based Student Health Surveys. Food Nutr Bull. 2019 Dec;40(4):444-459. doi: 10.1177/0379572119848287.].

Even CF teenagers are ... teenagers, and they have a certain propensity to rebellion. It would be interesting to evaluate if perception of unsatisfaction is age related!? Because if if is age related, Nerds candy could be an excellent option for increasing adherence to oral glucose tolerance screening tests in certain age sub-groups. 

3. Shared meals and dietary patterns are generating a contamination effect on food preferences in families. Availability of soft drinks and negative parental role modeling are important predictors of children's dietary preferences [Hebestreit A, et al. Dietary Patterns of European Children and Their  Parents in Association with Family Food  Environment: Results from the I.Family Study. Nutrients. 2017 Feb 10;9(2):126. doi: 10.3390/nu9020126.].

Were all these children "exposed" to similar familial eating patterns?

Your study has proven that carefully selected alternatives may produce comparable glycemic OGTT. Could you expand your paper with potential alternatives for Nerds candy, alternatives that could be evaluated in terms of tolerability in a future study?

Author Response

  1. A potential bias factor can be represented by socio-economic status, because such adolescents are presenting persistently unhealthier dietary profiles [Fernández-Alvira JM, et al. Prospective associations between socio-economic status and dietary patterns in European children: the Identification and Prevention of Dietary- and Lifestyle-induced Health Effects in Children and Infants (IDEFICS) Study. Br J Nutr. 2015 Feb 14;113(3):517-25. doi: 10.1017/S0007114514003663.].

Could you evaluate socio-economic distribution patterns and correlation in your small group? Are there any discrepancies in perception related to this aspect?

Reply:

Thank you for this thoughtful point on socioeconomic factors that contribute to overall outcomes. Though our patients all live within a close geographical radius, we did not do a formal analysis of socio-economic factors such as level of education of parents of patients, annual salary, access to resources such as food, etc. We did go back to our records to determine that out of the total cohort, 3 patients were covered by Medicaid, a public assistance program type insurance that is income-based and the remainder of patients were privately insured.

We have added this information as context and have addressed this as a limiting factor in the Limitations section from lines 176-180.  

  1. Adolescents have infrequently healthy eating habits, and this moment in time is a difficult one, during childhood [Beal T, Morris SS, Tumilowicz A. Global Patterns of Adolescent Fruit, Vegetable, Carbonated Soft Drink, and Fast-Food Consumption: A Meta-Analysis of Global School-Based Student Health Surveys. Food Nutr Bull. 2019 Dec;40(4):444-459. doi: 10.1177/0379572119848287.].

Even CF teenagers are ... teenagers, and they have a certain propensity to rebellion. It would be interesting to evaluate if perception of unsatisfaction is age related!? Because if is age related, Nerds candy could be an excellent option for increasing adherence to oral glucose tolerance screening tests in certain age sub-groups. 

Reply:

We agree that evaluating whether or not perception of Nerds vs ODS could be related to age of the patient. Unfortunately due to the size of our cohort during the time of the global pandemic we do not feel we have an adequate study sample to make this conclusion.

We have added the point beginning on line 142, “As studies have shown, age and stage of development has a correlation to food choices to preference and perception may be playing a larger role than realized” and have cited this paper as a reference.

  1. Shared meals and dietary patterns are generating a contamination effect on food preferences in families. Availability of soft drinks and negative parental role modeling are important predictors of children's dietary preferences [Hebestreit A, et al. Dietary Patterns of European Children and Their  Parents in Association with Family Food  Environment: Results from the I.Family Study. Nutrients. 2017 Feb 10;9(2):126. doi: 10.3390/nu9020126.].

Were all these children "exposed" to similar familial eating patterns?

Reply:

Thank you for this pertinent point that family eating patterns play a vital role in prediction of child dietary behaviors and preferences. We did not investigate background information such as family eating pattern (eg do all families eat at the dinner table together, do all members of the family aid in meal preparation, etc.) however we have acknowledged this as a limitation in the Limitations section beginning on line 180 and have cited this paper as a reference.

  1. Your study has proven that carefully selected alternatives may produce comparable glycemic OGTT. Could you expand your paper with potential alternatives for Nerds candy, alternatives that could be evaluated in terms of tolerability in a future study?

Reply:

We have expanded the study limitations starting at line 169 to discuss potential alternatives to NERDs candy.  Alternative candy could be used and important considerations would be choosing a local candy that does not contain ingredients prone to allergy or intolerance such as high fructose corn syrup, gluten, or gelatin products.  The candy choice should have no fat whatsoever and no protein.  In addition, the candy should not have any added types of fibers such as pectin. Lastly, a candy should be able to be in an easily weighable form to allow for careful weight-based dosing.  As discussed in our manuscript other studies have evaluated other potential candy alternatives for use in the OGTT. We felt our study was unique in that, to our knowledge, no other group had ever used Nerds candy. We agree that further research should be done with larger cohorts to evaluate tolerability of an expanded set of candy alternatives with hopes of improving adherence rates to the OGTT.

Round 2

Reviewer 1 Report

Edits nicely done.